# Personalized risk stratification through attribute matching for clinical decision making in clinical conditions with aspecific symptoms: The example of syncope

**Monica Solbiati**[1,2], **James V. Quinn**[3], **Franca Dipaola**[4], **Piergiorgio Duca**[5], **Raffaello Furlan**[4], **Nicola Montano**[2,6], **Matthew J. Reed** [7,8], **Robert S. Sheldon**[9], **Benjamin C. Sun**[10], **Andrea Ungar**[11], **Giovanni Casazza** [5]*, **Giorgio Costantino**[1,2], on behalf of the SYNERGI (SYNcope Expert Research Group International)[¶]

1 UOC Pronto Soccorso e Medicina d'Urgenza, Fondazione IRCCS Ca' Granda Ospedale Maggiore Policlinico, Milan, Italy, 2 Dipartimento di Scienze Cliniche e di Comunità, Università degli Studi di Milano, Milan, Italy, 3 Department of Emergency Medicine, Stanford University, Stanford, CA, United States of America, 4 Internal Medicine, Humanitas Research Hospital, Humanitas University, Rozzano, Italy, 5 Dipartimento di Scienze Biomediche e Cliniche "L. Sacco", Università degli Studi di Milano, Milan, Italy, 6 Dipartimento di Medicina Interna, Fondazione IRCCS Ca' Granda Ospedale Maggiore Policlinico, Milan, Italy, 7 Emergency Medicine Research Group Edinburgh (EMERGE), Royal Infirmary of Edinburgh, Edinburgh, United Kingdom, 8 Edinburgh Acute Care, Usher Institute of Population Health Sciences and Informatics, University of Edinburgh, Edinburgh, United Kingdom, 9 Libin Cardiovascular Institute of Alberta, University of Calgary, Calgary, Canada, 10 Department of Emergency Medicine, Center for Policy Research-Emergency Medicine, Oregon Health and Science University, Portland, OR, United States of America, 11 S. O.D. Geriatria e Terapia Intensiva Geriatrica, AOU Careggi e Università degli Studi di Firenze, Florence, Italy

¶ Membership of the SYNERGI (SYNcope Expert Research Group International) is provided in the Acknowledgments.
* giovanni.casazza@unimi.it

**Data Availability Statement:** Data contain potentially identifying or sensitive patient information. For this reason, data from the single

## Abstract

### Background

Risk stratification is challenging in conditions, such as chest pain, shortness of breath and syncope, which can be the manifestation of many possible underlying diseases. In these cases, decision tools are unlikely to accurately identify all the different adverse events related to the possible etiologies. Attribute matching is a prediction method that matches an individual patient to a group of previously observed patients with identical characteristics and known outcome. We used syncope as a paradigm of clinical conditions presenting with aspecific symptoms to test the attribute matching method for the prediction of the personalized risk of adverse events.

### Methods

We selected the 8 predictor variables common to the individual-patient dataset of 5 prospective emergency department studies enrolling 3388 syncope patients. We calculated all possible combinations and the number of patients in each combination. We compared the predictive accuracy of attribute matching and logistic regression. We then classified ten random patients according to clinical judgment and attribute matching.

datasets included in this study cannot be shared publicly and are available upon request. Data are available from the single studies sites for researchers who meet the criteria for access to confidential data. Data requests may be sent to: sincope@policlinico.mi.it.

**Funding:** The authors received no specific funding for this work.

**Competing interests:** The authors have declared that no competing interests exist.

## Results

Attribute matching provided 253 of the 384 possible combinations in the dataset. Twelve (4.7%), 35 (13.8%), 50 (19.8%) and 160 (63.2%) combinations had a match size $\geq$50, $\geq$30, $\geq$20 and <10 patients, respectively. The AUC for the attribute matching and the multivariate model were 0.59 and 0.74, respectively.

## Conclusions

Attribute matching is a promising tool for personalized and flexible risk prediction. Large databases will need to be used in future studies to test and apply the method in different conditions.

## Introduction

Clinical decision tools (CDT) combine different predictors (from patients' history, clinical examination and tests results) to assess the probability of a diagnosis, prognosis, or response to treatment of an individual patient [1]. The statistical techniques used in this process are usually based on multivariate models such as logistic regression [2]. Other approaches include recursive partitioning analysis and artificial neural networks [3–5]. As they are based on models, CDTs are able to predict the risk of any hypothetical patient, even those with a combination of risk factors different from all the patients of the derivation cohort. Therefore, we do not know how the CDT will perform in subjects with specific clinical presentations or needs. Indeed, they lack the ability to provide personalized estimates as required in the era of precision medicine. For example, patients with uncommon diseases are likely not to be correctly risk stratified by CDTs. In addition, the risk estimates of composite outcomes that are usually provided by CDTs cannot always be applied to all patients, as the definition of "acceptable risk" depends on the patient at risk. Hence the need to assess a personalized risk rather than providing a simple binary answer [6].

Moreover, risk stratification is challenging in conditions (as chest pain, shortness of breath and syncope) presenting with aspecific symptoms that can be the manifestation of many possible underlying diseases. In these cases, decision tools are unlikely to accurately identify all the different adverse events related to the possible etiologies. In syncope, which is a paradigm of the above conditions, the traditionally derived risk stratification tools have failed in predicting adverse events [7–12]. Here, an individualized risk assessment would allow an estimate of not only the probability of a composite endpoint, but rather a detailed risk profile that provides the individual risk of each specific outcome (e.g. arrhythmia or pulmonary embolism).

Attribute matching (AM) is a prediction approach that differs considerably from the regression models and has shown promising results in ruling out acute coronary syndrome and pulmonary embolism in patients with chest pain [13–15]. Instead of considering each clinical characteristic as an individual predictor and deriving a risk estimate based on the sum of their regression coefficients, each individual patient is matched to a group of patients with the same combination of the relevant clinical characteristics (or attributes) from a large reference database. Therefore, each patient is matched to a group of patients with identical risk profile and known outcomes. This approach results in a proportion (i.e. the number patients who had the outcome of interest on the number of previously studied matched patients) that provides the probability (with confidence interval) of the single adverse event. This process resembles the

definition of pre-test probability by an expert clinician, which, having seen many patients who had similar clinical characteristics as the patient under consideration, could provide an estimate of the probability of something bad happening. In this case, the computer does so with less variability and without the clinician having to be experienced nor an expert. The aim of this study was to explore the use of AM to predict the personalized risk of adverse events and to compare it to multivariate logistic regression to analyze the possible similarities, differences, strengths and weaknesses of the two methods using syncope in the Emergency Department (ED) as an example.

## Materials and methods

To apply AM in a large database, we used an individual-patient dataset from a previous international collaboration that involved 3388 patients prospectively included in 5 studies enrolling syncope patients in the ED from 2000 to 2014 [8,16–20]. The dataset was analyzed to detect demographic and clinical variables among those considered to be relevant for syncope risk stratification as have shown to be related to adverse events [16,17,19,21]. Each single dataset was re-analyzed to create homogeneously defined variables for abnormal electrocardiogram (ECG) and 7–10 day serious outcomes [7,12,22]. We finally identified the variables that were available in all 5 datasets.

The AM estimates of the probability of serious adverse is based upon computer assisted, database-derived system. The clinician puts in a predefined set of clinical attributes for a subject for whom the probability of a serious outcome is unknown. A computer program queries a large patient database, and returns only the patients who share the identical attribute profile as the patient under consideration. The proportion of these attribute-matched subjects who had a clinical outcome of interest is the probability of adverse events.

According to the "Standardized reporting guidelines for emergency department syncope risk-stratification research" serious outcomes included any of the following [22]: 1) all-cause and syncope-related death, 2) ventricular fibrillation, 3) sustained and symptomatic non-sustained ventricular tachycardia, 4) sinus arrest with cardiac pause > 3 s, 5) sick sinus syndrome with alternating bradycardia and tachycardia, 6) second-degree type 2 or third-degree AV block, 7) permanent pacemaker (PM) or implantable cardioverter defibrillator (ICD) malfunction with cardiac pauses, 8) aortic stenosis with valve area $\leq$ 1 cm2, 9) hypertrophic cardiomyopathy with outflow tract obstruction, 10) left atrial myxoma or thrombus with outflow tract obstruction, 11) myocardial infarction, 12) pulmonary embolism, 13) aortic dissection, 14) occult hemorrhage or anemia requiring transfusion, 15) syncope or fall resulting in major traumatic injury (requiring admission or procedural/surgical intervention), 16) PM or ICD implantation, 17) cardiopulmonary resuscitation, 18) syncope recurrence with hospital admission, and 19) cerebrovascular events.

To explore the potential application of AM in this context, we calculated 1) all the unique combinations of the selected variables (or attributes); 2) the number of combinations verified in at least one patient in the database; 3) the number of combinations with a match size $\geq$50, $\geq$ 30, $\geq$20 and <10 patients.

The potential predictors of short-term severe outcomes were first individually evaluated and then analyzed by multivariate logistic regression analysis with a stepwise selection strategy. In case of one predictor was missing in one patient, it was considered as absent.

The overall diagnostic performance of both multivariate logistic regression and AM was assessed with Receiver Operating Characteristics (ROC) curves and their area under the curve (AUC).To exemplify how the AM would work in the real world, we considered 10 random patients who presented with syncope, as defined according to the main international

guidelines and consensus papers [11,12], to the ED of Fondazione IRCCS Ca' Granda, Ospedale Maggiore Policlinico, Milano from September 2015 to February 2017 [23]. For each patient we recorded the presence or absence of any of the above attributes and calculated the risk of adverse events according to the AM approach. For this purpose we paired the patient of interest to the patients with an identical combination of attributes in the database and calculated the probability of adverse events as the percentage of the matched previously studied patients who had the outcome of interest [13]. A 95% confidence interval (CI) was constructed using the binomial distribution. As part of a larger study on syncope ED risk stratification, we asked the ED physician to assess the patient's risk of short-term adverse events (low, intermediate or high) according to his/her clinical judgement.

The data for this study were collected and analyzed anonymously. The 10 random example patients had given written informed to have their data collected and the Internal Review Board of L. Sacco Hospital (approval number 608/2015) had approved their use for this study purpose. IRB approval was obtained by the single primary study authors.

Analyses were performed using the SAS (release 9.4) statistical software.

## Results

The main characteristics of the 3388 patients included in the individual-patient database are reported in Table 1. We identified 8 common predictors: sex, age (considered as a 3-level categorical variable: < 45 year, ≥ 45 and < 65 years, ≥ 65 years), trauma following syncope, presence of abnormal ECG, history of cerebrovascular disease, history of cardiac disease, history of syncope and absence of prodrome.

**Table 1. Characteristics of the included patients.**

| Variables | EGSYS [18,24] | SFSR [19] | STePS [16] | ROSE [17] | Sun 2007 [20] | Total |
|---|---|---|---|---|---|---|
| Total number of patients | 465 | 684 | 695 | 1067 | 477 | 3388 |
| Age, median (IQR) | 70 (45–81) | 70 (42–81) | 64 (41–78) | 69 (48–81) | 58 (35–79) | 67 (43–80) |
| N of admitted patients (%) | 178 (38) | 364 (53) | 265 (38) | 538 (50) | 286 (60) | 1631 (48) |
| N of men (%) | 253 (54) | 281 (41) | 306 (44) | 480 (45) | 210 (44) | 1530 (45) |
| N of patients with history of syncope (%) | 195 (42) | 124 (18) | 389 (56) | 176 (16) | 160/457 (34) | 1044/2931 (36) |
| N of patients without prodrome (%) | 122 (26) | 260 (38) | 195 (28) | 410 (38) | 141 (30) | 1128 (33) |
| N of patients with trauma following syncope (%) | 133 (29) | 45 (7) | 162 (23) | 316 (30) | n.a. | 656/2911 (23) |
| N of patients with abnormal ECG (%) | 178 (38) | 222 (32) | 202 (29) | 665 (62) | 170 (36) | 1437 (42) |
| N of patients with a history of cardiovascular disease (%) | 153 (33) | 139 (20) | 178 (26) | 284 (27) | 150 (31) | 904 (27) |
| N of patients with a history of cerebrovascular disease (%) | 166 (36) | 115 (17) | 227 (33) | n.a. | 169 (35) | 677/2321 (29) |
| N of patients with serious outcomes at 10 days (%)* | 93 (20) | 81 (12) | 44 (6) | 49 (5) | 62 (13) | 329 (10) |
| N of deaths | 6 | 6 | 7 | 6 | 1 | 26 (1) |
| N of arrhythmias | 31 | 30 | | 20 | 32 | |
| N of cardiopulmonary resuscitations | | | 5 | 2 | | |
| N of myocardial infarctions | 6 | 33 | | | 1 | |
| N of structural cardiopulmonary diseases | 9 | 10 | | 14 | 6 | |
| N of PM insertions or malfunctions | 43 | | 25 | 11 | 2 | |
| N of ICD insertions or malfunctions | 5 | | 2 | | | |
| N of haemorrhages | | 24 | | 7 | 8 | |

IQR: interquartile range; ECG: electrocardiogram; PM: pacemaker; ICD: Implantable Cardioverter Defibrillator; n.a.: not available.

*Some patients had more than one outcome.

**Table 2. Risk factors for severe short-term outcomes within 10 days (univariate analysis).**

| | Severe Outcomes | | |
|---|---|---|---|
| | Yes (%) (n = 329) | No (%) (n = 3059) | p-value* |
| Male gender, n (%) | 196 (60) | 1334 (44) | <0.0001 |
| Age, n (%) | | | <0.0001 |
| < 45 years | 24 (7) | 869 (28) | |
| ≥ 45 and < 65 years | 56 (17) | 658 (22) | |
| ≥ 65 years | 249 (76) | 1532 (50) | |
| Syncope during exertion, n (%) | 31 (9) | 187 (6) | 0.0211 |
| Trauma following syncope, n (%) | 64 (19) | 592 (19) | 0.9651 |
| Abnormal ECG, n (%) | 229 (70) | 1208 (39) | <0.0001 |
| Medical history, n (%) | | | |
| Cardiovascular disease | 161 (49) | 743 (24) | <0.0001 |
| Cerebrovascular disease | 132 (40) | 545 (18) | <0.0001 |
| Arterial hypertension | 154 (47) | 1104 (36) | 0.0001 |
| Previous syncope | 109 (33) | 964 (31) | 0.5491 |
| Absence of prodrome, n (%) | 126 (38) | 1002 (33) | 0.0430 |

*Chi-square test; ECG: electrocardiogram

The AM method provided 253 of the 384 possible combinations. No patient in the database matched the remaining 131 combinations of predictors. Only 12 of the 253 (4.7%) combinations had a match size ≥50 patients, 35 (13.8%) had a match size ≥30 patients, 50 (19.8%) had a match size ≥20 patients, and most (160, 63.2%) had a match size <10 patients.

At univariate analysis, the risk factors significantly associated with severe short-term outcomes were age, male gender, syncope during exertion, abnormal ECG, history of cardiovascular disease, history of cerebrovascular disease, absence of prodrome, and history of arterial hypertension (Table 2).

At multivariate analysis, male gender, age between 45 and 65 years, age over 65 years, an abnormal ECG, and a past medical history of cerebrovascular disease were independent risk factors for the development of severe adverse outcomes in the short term (Table 3).

The AUC for the AM and the multivariate model were 0.59 and 0.74, respectively.

The predicted probabilities for each of the 10 patients, together with the ED physician's perceived risk are reported in Table 4. To note, none of these patients had an adverse event at 7–30 days of follow-up according to standardized criteria [22]. The detailed case description of the 10 patients is reported in S1 Table.

**Table 3. Risk factors for severe short-term outcomes within 10 days at logistic multivariate regression (stepwise selection).**

| | Adjusted Odds Ratio | 95% Confidence Interval | p-value* |
|---|---|---|---|
| Male gender | 1.6 | 1.3–2.0 | 0.0001 |
| Age | | | <0.0001 |
| < 45 years | 1.0 | | |
| ≥ 45 and < 65 years | 2.3 | 1.4–3.8 | |
| ≥ 65 years | 3.5 | 2.3–5.5 | |
| Abnormal ECG | 2.6 | 2.0–3.3 | <0.0001 |
| Medical history of cerebrovascular disease | 1.9 | 1.5–2.5 | <0.0001 |

*Chi-square test

ECG: electrocardiogram

**Table 4. Predicted probabilities according to attribute matching and clinical judgement in the 10 example patients.**

| Case n | Attribute matching | | ED physician |
|---|---|---|---|
| | patients at risk* | 10-day SAE, % (95% CI) | |
| 1 | 15 | 20 (7–45) | High risk |
| 2 | 70 | 4 (1–12) | Intermediate risk |
| 3 | 42 | 5 (1–16) | Intermediate risk |
| 4 | 12 | 0 (0–24) | Intermediate risk |
| 5 | 84 | 4 (1–10) | Intermediate risk |
| 6 | 34 | 6 (2–19) | Low risk |
| 7 | 42 | 5 (1–16) | High risk |
| 8 | 6 | 16 (3–56) | High risk |
| 9 | 6 | 0 (0–39) | High risk |
| 10 | 3 | 33 (6–79) | High risk |

ED: Emergency Department; SAE: serious adverse events

*: number of patients with the same combination of risk factors

CI: Confidence Interval.

## Discussion

In this paper, to assess the potential value of AM and to compare it to multivariate logistic regression we used syncope as a paradigm of those conditions, such as chest pain and shortness of breath, in which the creation of accurate CDTs is particularly challenging. If the condition under consideration is the manifestation of many possible underlying diseases, CDTs are unlikely to accurately identify all the different adverse events related to the possible etiologies [25]. In syncope, CDTs are usually designed to identify multiple diagnoses (i.e. pulmonary embolism, aortic dissection, high grade atrioventricular block) and adverse events that might be related to a high number of conditions (i.e. bleeding requiring transfusion, trauma, pacemaker implant). To increase complexity, the reference standard for diagnosis is sometimes missing.

This study explores a method to estimate the probability of serious adverse events based on AM. This approach allows the clinician to determine the probability of a serious outcome of a patient based on the presence of predefined risk predictors (or attributes). This patient is matched to all patients with the same combination of attributes included in a large reference database. The proportion of these attribute-matched patients who had the outcome of interest represents the estimate, with its 95% confidence interval, of the probability that such outcome might occur in the patient under consideration [15]. This process resembles the definition of pre-test probability by an expert clinician, which, having seen many patients who had similar clinical characteristics as the patient under consideration, could provide an estimate of the probability of something bad happening. In this case the computer does so with less variability and without the clinician having to be experienced nor an expert.

The inclusion of a large number of attributes would result in very specific and detailed clinical risk profiles at a cost of requiring a very large reference database. In the present work, we used an eight-attribute profile and a 3388-patient database. Among the 384 possible combinations, only 12 had a match size ≥50 patients and most had a match size <10 patients. Therefore, our data do not offer a clinically useful prediction tool at this stage and the AUC shows that logistic regression is superior if derived from the dataset we used, but this method seems promising, as it has some advantages as compared to model-derived clinical decision tools.

Indeed, the successful use of a model to predict the probability of a serious outcome requires that the results are reproduced in an external validation so that both the external validity and robustness of the model are verified. Moreover, models require that the predictors are assigned a weight that allow to estimate the risk of adverse events in every patient, also in those that had no matching subject in the derivation database (for example for patients that have a rare condition). Attribute matching differs from scoring systems derived from logistic regression, which use predictor variables expressed by an individual patient under consideration to guide that patient into a predefined category that predicts a probability. This outcome probability is estimated from knowledge (i.e., the magnitude of importance of predictor variables) manifested by the patients that were used to construct the model. On the other hand, attribute matching works in reverse fashion. Instead of placing the patient under consideration into a category, the computer program finds the patients from a reference database who "look like" the patient insofar as they are identical on the binary predictor variables. Therefore, the risk of patients with an uncommon combination of predictors, might not be able at all to find a match in the derivation dataset. However, being aware that the patient's estimated probability might be based on very limited evidence, will allow both the clinician and the patient to take a decision conscious that it might be based on uncertainty, rather than deciding on the false confidence provided by models.

Several thousands of subjects need to be enrolled for acceptable AM risk prediction. If this was the case, only administrative databases could be used to use AM for risk prediction. In the era of big data and with the increase in the availability and accuracy of population-based databases, this might not be a barrier to the use of AM for risk prediction in several conditions [26].

AM has several advantages: 1) The possibility to have as output not only the probability of a composite serious outcome, but a detailed patient specific risk profile based on the probability of different outcomes allowing for a more personalized decision making. Also, the possibility to make the risk profile explicit and more personalized could allow for more meaningful shared decision making with the patient; 2) as there is no need for model fitting, patients could be always added to the dataset thus increasing the probability estimate precision; 3) the flexibility of AM would allow to consider different predictors in different patients, thus allowing an individualized estimate; 4) as there is no statistical modelling, the reliability of the results is based on the similarity between the population of the reference database and every-day patients rather than on complex statistical calculations; 5) the prediction tools based on models, such as logistic regression and neural networks provide a risk estimate in every case, also in patients whose combination of clinical characteristics are different from each patient's combination in the derivation cohort, giving the physician a false confidence. Conversely, AM would allow both the clinician and the patient to make a decision being aware that it might be based on uncertainty, rather than deciding on the false confidence provided by models. This is crucial in the perspective of a modern medicine increasingly based on personalized and shared decision making.

AM has also some important limitations: 1) to be used in clinical practice the reference database should include a large number of patients; 2) the choice of predictors is crucial for the successful application of the method; 3) AM will promote personalized medicine, providing the probability of events, rather than a clear indication of what to do (i.e. admit vs discharge). However, the need to interpret and apply the estimated probability to the context may be felt as a limitation due to lack of certainty; 4) a score is easy to remember and apply, while AM requires data collection and computer input ideally through a computer/smartphone app. Furthermore, the value of CDT as early and necessary work to determine the choice of predictors to be considered should not be under estimated as they help determine what attributes and factors should be collected and used for AM.

Some limitations of the present study should be acknowledged. The database we used was collected for different purposes and, although we did our best to homogenize the data, we could not overcome some heterogeneity among the single studies' dataset. Also, we used as predictors the eight variables in common between the original datasets with no *a priori* decision on the number of predictors to be selected. However, this number strongly influences the sample size of the population to be included in the AM database. Nonetheless, it must be pointed out that syncope and this database were used only as a working example to show the possible applications of AM.

## Conclusions

In conclusion, our study shows that the AM is a promising method to predict the risk of adverse events in clinical practice and could offer some advantages as compared with standard methods based on logistic regression. However, large datasets are required to obtain a precise and informative estimate. Future studies should explore the use of administrative databased or big data in conditions in which there is less clinical heterogeneity to use AM and to compare it with the traditional risk stratification tools.

## Supporting information

**S1 Table. Example clinical cases with the probabilities predicted by attribute matching and clinical judgement.** BP: blood pressure; HR: heart rate; ECG: electrocardiogram; ED: Emergency Department; CI: Confidence Interval.
(DOCX)

## Acknowledgments

Members of the SYNcope Expert Research Group International (SYNERGI):

- Franca Barbic: Internal Medicine, Humanitas Research Hospital, Humanitas University, Rozzano, Italy

- Giovanni Casazza: Dipartimento di Scienze Biomediche e Cliniche "L. Sacco", Università degli Studi di Milano, Milan, Italy

- Giorgio Costantino (lead author): UOC Pronto Soccorso e Medicina d'Urgenza, Fondazione IRCCS Ca' Granda, Ospedale Maggiore Policlinico, and Dipartimento di Scienze Cliniche e di Comunità, Università degli Studi di Milano, Milan, Italy, giorgic2@gmail.com

- Franca Dipaola: Internal Medicine, Humanitas Research Hospital, Humanitas University, Rozzano, Italy

- Raffaello Furlan: Internal Medicine, Humanitas Research Hospital, Humanitas University, Rozzano, Italy

- Rose A Kenny: Department of Medical Gerontology, Trinity College, Dublin, Ireland

- James V Quinn: Department of Emergency Medicine, Stanford University, Stanford, CA, USA

- Satish R Raj: Libin Cardiovascular Institute of Alberta, University of Calgary, Calgary, Canada

• Matthew J Reed: Emergency Medicine Research Group Edinburgh (EMERGE), Royal Infirmary of Edinburgh, and Edinburgh Acute Care, Usher Institute of Population Health Sciences and Informatics, University of Edinburgh, Edinburgh, UK

• Robert S Sheldon: Libin Cardiovascular Institute of Alberta, University of Calgary, Calgary, Canada

• Win-Kuang Shen: Department of Cardiovascular Medicine, Mayo Clinic, Phoenix, AZ, USA

• Monica Solbiati: UOC Pronto Soccorso e Medicina d'Urgenza, Fondazione IRCCS Ca' Granda, Ospedale Maggiore Policlinico, and Dipartimento di Scienze Cliniche e di Comunità, Università degli Studi di Milano, Milan, Italy

• Benjamin C. Sun: Department of Emergency Medicine, Center for Policy Research-Emergency Medicine, Oregon Health and Science University, Portland, OR, USA

• Venkatesh Thiruganasambandamoorthy: Department of Emergency Medicine, University of Ottawa, Ottawa, ON, Canada

## Author Contributions

**Conceptualization:** Monica Solbiati, James V. Quinn, Giovanni Casazza, Giorgio Costantino.

**Data curation:** Monica Solbiati, Giovanni Casazza, Giorgio Costantino.

**Formal analysis:** Monica Solbiati, Giovanni Casazza, Giorgio Costantino.

**Investigation:** Monica Solbiati, James V. Quinn, Giovanni Casazza, Giorgio Costantino.

**Methodology:** Monica Solbiati, Giovanni Casazza, Giorgio Costantino.

**Project administration:** Monica Solbiati, Giovanni Casazza, Giorgio Costantino.

**Supervision:** Monica Solbiati, Giovanni Casazza, Giorgio Costantino.

**Writing – original draft:** Monica Solbiati, Giovanni Casazza, Giorgio Costantino.

**Writing – review & editing:** James V. Quinn, Franca Dipaola, Piergiorgio Duca, Raffaello Furlan, Nicola Montano, Matthew J. Reed, Robert S. Sheldon, Benjamin C. Sun, Andrea Ungar.

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
