## [Editor Report · Decision Letter 0]

20 Sep 2019

PONE-D-19-19894

Personalized risk stratification through attribute matching using syncope as a clinical example

PLOS ONE

Dear Dr Casazza

Thank you for submitting your manuscript to PLOS ONE. After careful consideration, we feel that it has merit but does not fully meet PLOS ONE’s publication criteria as it currently stands. Therefore, we invite you to submit a revised version of the manuscript that addresses the points raised during the review process.

We would appreciate receiving your revised manuscript in 60 days. To enhance the reproducibility of your results, we recommend that if applicable you deposit your laboratory protocols in protocols.io, where a protocol can be assigned its own identifier (DOI) such that it can be cited independently in the future. For instructions see: http://journals.plos.org/plosone/s/submission-guidelines#loc-laboratory-protocols

We look forward to receiving your revised manuscript.

Kind regards,

Sandro Pasquali, M.D., Ph.D.

Academic Editor

PLOS ONE

This is a good study with some interestign findings. However it is poorly decribed and reported. The Authors should make any effort to increase the quality of their manuscript. Here are some issues to be addressed.

The study is aimed at investigating a new methodology rather than a clinical condition. The title should be reflect the study aim and reworded. For instance, a possibility is “Personalized risk stratification through attribute matching for clinical decision making in clinical conditions with aspecific symptoms: the example of syncope”.

The clinical challenge of predicting patient risk in an aspecific condition is well presented in the letter to the Editor. However this remains quite unclear in the abstract. It should be made clear in the abstract that the Authors are investigating a promising methodology for quite generic symptoms and that they picked up syncope as an example.

Introduction. The sentence “While CDTs provide information that would be applicable to the nonspecific patient, they lack useful and precise prediction in subjects with specific clinical presentations or needs.” Is probably inaccurate. Prediction tools can be accurate also on a specicific patient, they just need to be informative in terms of variables. For instance CTDs are sometimes very accurate in cancer medicine and they are replacing AJCC TNM staging system as highlighted in the 8th edition of the TMN staging manual.

Introduction. Introduction is also likely missing the point. The Authors stated that “Since the well-known limitations of the traditionally derived risk stratification tools in predicting adverse events after syncope”. Are the Authors looking at predicting the cause of a condition that underlie a symptom, that is here syncope?

Introduction. It should be stated which is the difference between standard predicting tools and attribute matching. The description about AM is in the method but it seems more appropriate to move it to the Intro to facilitate the reading.

Introduction. Study hypothesis and aims should be clearly stated. It should be stated that the Authors compared AM prediction and pre-probability defined by an expert clinican. This is quite interesting as the Authors probably expected that a large dataset works as accurately as an expert clinicians though in a more reporducible way when compared to a clinician.

Introduction. AM should be spelled out.

Methods. Time frame of the study is needed.

Methods. Statistical methodology for AM should be biefly reported and referenced.

Methods. Outcome variables and measures not reported.

Results. Which are the 8 selected variables?

Results. ED physician. This should be reported in the method section and criteria for each category described.

Methods/results. The authors mentioned that AM works better than traditional prediction, which may well be the case. However, in the manuscript such comparison has not been made. Since the Authors have a pretty large dataset they should be able to run this comparison. For instance, they can fit a predictive model based on regression for predicting SAE and test in on their 10 prospective patients. Then they should compare AM and traditional prediction based on regression.

Method/results. Are 10 patients enough for this study? Was a sample size calculation performed? If not, which is the reason?

Discussion. This is a well balanced dìscussion.

2. Thank you for including your ethics statement:  "The data for this study were collected and analyzed anonymously.

The 10 patients in Table 2 had given written informed to have their data collected and the Internal Review Board had approved their use for this study purpose.

IRB approval was obtained by the single primary study authors.".   

a.Please amend your current ethics statement to include the full name of the ethics committee/institutional review board(s) that approved your specific study.

b.Once you have amended this/these statement(s) in the Methods section of the manuscript, please add the same text to the “Ethics Statement” field of the submission form (via “Edit Submission”).

4. One of the noted authors is a group or consortium SYNERGI (SYNcope Expert Research Group International). In addition to naming the author group, please list the individual authors and affiliations within this group in the acknowledgments section of your manuscript. Please also indicate clearly a lead author for this group along with a contact email address.
---

## [Author Response · Author response to Decision Letter 0]

3 Nov 2019

Dear Dr. Pasquali, 

Thank you for having considered our manuscript entitled “Personalized risk stratification through attribute matching using syncope as a clinical example” for publication in PLOS ONE (PONE-D-19-19894). We would also like to thank you for the thoughtful comments: we have revised the manuscript accordingly and we believe that this process has improved its quality.

Please find attached a marked-up copy of the manuscript that highlights changes made to the original version and an unmarked version of the revised paper without tracked changes, together with point-by-point responses to the points raised during the review process. 

Sincerely yours,

Giovanni Casazza, on behalf of all Co-Authors

Here are our responses to the specific points raised during the review process.

This is a good study with some interesting findings. However it is poorly decribed and reported. The Authors should make any effort to increase the quality of their manuscript. Here are some issues to be addressed.

The study is aimed at investigating a new methodology rather than a clinical condition. The title should be reflect the study aim and reworded. For instance, a possibility is “Personalized risk stratification through attribute matching for clinical decision making in clinical conditions with aspecific symptoms: the example of syncope”.

We appreciate the editor’s suggestion. We changed the title as suggested.

The clinical challenge of predicting patient risk in an aspecific condition is well presented in the letter to the Editor. However this remains quite unclear in the abstract. It should be made clear in the abstract that the Authors are investigating a promising methodology for quite generic symptoms and that they picked up syncope as an example.

We appreciate the opportunity to clarify. We changed the abstract accordingly.

Introduction. The sentence “While CDTs provide information that would be applicable to the nonspecific patient, they lack useful and precise prediction in subjects with specific clinical presentations or needs.” Is probably inaccurate. Prediction tools can be accurate also on a specific patient, they just need to be informative in terms of variables. For instance CTDs are sometimes very accurate in cancer medicine and they are replacing AJCC TNM staging system as highlighted in the 8th edition of the TMN staging manual.

We agree with the editor that this statement was inaccurate. We tried to explain better our point of view. The sentence now is the following: “As they are based on models, CDTs are able to predict the risk of any hypothetical patient, even those with a combination of risk factors different from all the patients of the derivation cohort. Therefore, we do not know how the CDT will perform in subjects with specific clinical presentations or needs. Indeed, they lack the ability to provide personalized estimates as required in the era of precision medicine. For example, patients with uncommon diseases are likely not to be correctly risk stratified by CDTs. In addition, the risk estimates of composite outcomes that are usually provided by CDTs cannot always be applied to all patients, as the definition of “acceptable risk” depends on the patient at risk. Hence the need to assess a personalized risk rather than providing a simple binary answer [6].”

Introduction. Introduction is also likely missing the point. The Authors stated that “Since the well-known limitations of the traditionally derived risk stratification tools in predicting adverse events after syncope”. Are the Authors looking at predicting the cause of a condition that underlie a symptom, that is here syncope?

We appreciate the opportunity to clarify. We changed the text as follows: Moreover, risk stratification is challenging in conditions (as chest pain, shortness of breath and syncope) presenting with aspecific symptoms that can be the manifestation of many possible underlying diseases. In these cases, decision tools are unlikely to accurately identify all the different adverse events related to the possible etiologies. In syncope, which is a paradigm of the above conditions, the traditionally derived risk stratification tools have failed in predicting adverse events [7–12]. Here, an individualized risk assessment would allow an estimate of not only the probability of a composite endpoint, but rather a detailed risk profile that provides the individual risk of each specific outcome (e.g. arrhythmia or pulmonary embolism).”

Introduction. It should be stated which is the difference between standard predicting tools and attribute matching. The description about AM is in the method but it seems more appropriate to move it to the Intro to facilitate the reading.

We appreciate the editor’s suggestion. We moved the description about the AM method and the difference between standard predicting tools and AM in the introduction.

Introduction. Study hypothesis and aims should be clearly stated. It should be stated that the Authors compared AM prediction and pre-probability defined by an expert clinican. This is quite interesting as the Authors probably expected that a large dataset works as accurately as an expert clinicians though in a more reporducible way when compared to a clinician.

We appreciate the opportunity to clarify. In the present study, we did not perform a formal comparison between AM and clinical judgement. Indeed, the purpose of the study was to describe how the AM method could work in conditions like syncope. A comparison between AM and clinical judgement would require a much larger reference dataset. We just reported some examples of how it could work in real practice. We tried to explain this in the last paragraph of the methods section.

Introduction. AM should be spelled out.

We appreciate the editor’s suggestion. We added spelled out AM as suggested. 

Methods. Time frame of the study is needed.

We appreciate the editor’s suggestion. We added the study time-frame as required. 

Methods. Statistical methodology for AM should be biefly reported and referenced.

We appreciate the opportunity to clarify. We now added some details on the AM methodology and a reference in the methods section.

Methods. Outcome variables and measures not reported.

We appreciate the opportunity to clarify. We added the outcomes of interest in the methods section.

Results. Which are the 8 selected variables?

We appreciate the editor’s suggestion. We moved the list of the 8 selected variables from the methods to the results section.

Results. ED physician. This should be reported in the method section and criteria for each category described.

We appreciate the editor’s suggestion. We reported the ED physician role in the methods section. As described, there were no criteria to assess risk, as this was left to the ED physician’s judgement. 

Methods/results. The authors mentioned that AM works better than traditional prediction, which may well be the case. However, in the manuscript such comparison has not been made. Since the Authors have a pretty large dataset they should be able to run this comparison. For instance, they can fit a predictive model based on regression for predicting SAE and test in on their 10 prospective patients. Then they should compare AM and traditional prediction based on regression.

We appreciate the opportunity to clarify. The purpose of this study was not to make a comparison between AM and the traditional prediction methods. Indeed, we cannot state that AM works better, because a formal comparison would require thousands of patients in the reference database for AM to work. As stated in the study aim, we only wanted to test how AM could work with a real life example and to show that it could allow a different approach.

Method/results. Are 10 patients enough for this study? Was a sample size calculation performed? If not, which is the reason?

We would like to thank the editor for this comment. As the 10 patients were only an example and we did no attempt to “validate” the AM method on them, no sample size calculation was performed. If the editor feels that this in confusing rather than useful, we could remove the 10 example patients from the manuscript.

Discussion. This is a well balanced discussion.

We appreciate the editor’s comment

---

## [Decision Letter · Decision Letter 1]

21 Nov 2019

PONE-D-19-19894R1

Personalized risk stratification through attribute matching for clinical decision making in clinical conditions with aspecific symptoms: the example of syncope

PLOS ONE

Dear Prof Casazza,

Thank you for submitting your manuscript to PLOS ONE. After careful consideration, we feel that it has merit but does not fully meet PLOS ONE’s publication criteria as it currently stands. Therefore, we invite you to submit a revised version of the manuscript that addresses the points raised during the review process.

We would appreciate receiving your revised manuscript by Jan 05 2020 11:59PM. To enhance the reproducibility of your results, we recommend that if applicable you deposit your laboratory protocols in protocols.io, where a protocol can be assigned its own identifier (DOI) such that it can be cited independently in the future. For instructions see: http://journals.plos.org/plosone/s/submission-guidelines#loc-laboratory-protocols

We look forward to receiving your revised manuscript.

Kind regards,

Sandro Pasquali, M.D., Ph.D.

Academic Editor

PLOS ONE

Additional Editor Comments (if provided):

Authors have replied to comments raised in the previous review. Although they agreed in principle with the comments, only minor changes have been made throughout the text. More substantial changes are needed to improve the manuscript. The main idea behind this manuscript should be that the presented methodlogy is interesting and promising and a pilot has been conducted to show this. A larger study is clearly needed to validate the method, either looking at syncope or other conditions. In other words, findings are hypothesis generating rather than conclusive. In this regards, authors should make very clear what their next step will be.

The manuscript has been sent for additional review and comments of Reviewer#2 which are now available need to be carefully addressed in order to meet requirement for publication in PLOS ONE.

Reviewers' comments:

Reviewer's Responses to Questions

**Comments to the Author**

1. If the authors have adequately addressed your comments raised in a previous round of review and you feel that this manuscript is now acceptable for publication, you may indicate that here to bypass the “Comments to the Author” section, enter your conflict of interest statement in the “Confidential to Editor” section, and submit your "Accept" recommendation.

Reviewer #1: All comments have been addressed

2. Is the manuscript technically sound, and do the data support the conclusions?

Reviewer #1: Partly

3. Has the statistical analysis been performed appropriately and rigorously? 

Reviewer #1: No

4. Have the authors made all data underlying the findings in their manuscript fully available?

Reviewer #1: No

5. Is the manuscript presented in an intelligible fashion and written in standard English?

Reviewer #1: Yes

6. Review Comments to the Author

Reviewer #1: The authors quote the following sentence in the conclusions: "our study shows that the AM method could be used to predict the risk of adverse events in clinical practice". However, the paper makes no systematic comparison with the state of the art to support this view. Without this comparison the paper turns out to be nothing more than an exercise in style.

Several times the authors have been asked for this by the reviewers, with an answer "we did not perform a formal comparison

between AM and clinical judgement. Indeed, the purpose of the study was to describe how the AM method could work in conditions like syncope" and "The purpose of this study was not to make a comparison between AM and the traditional prediction methods. Indeed, we cannot state that AM works better, because a formal comparison would require thousands of patients in the reference database for AM to work. As stated in the study aim, we only wanted to test how AM could work with a real life example and to show that it could allow a different approach". These answers, from my personal point of view, indicate a lack of attention in the normal scientific approach in which, if a method is to be shown, all the specifications for comparison with the other methods must also be provided.

Please, provide:

- variable selection criterion and cut-points justification

- prediction accuracy measure, in order to enable the reader to judge the method.

7. PLOS authors have the option to publish the peer review history of their article (what does this mean?). If published, this will include your full peer review and any attached files.

Reviewer #1: No

---

## [Author Response · Author response to Decision Letter 1]

5 Jan 2020

Sandro Pasquali, M.D., Ph.D.

Academic Editor - PLOS ONE

Dear Dr. Pasquali, 

Thank you for having considered our revised manuscript entitled “Personalized risk stratification through attribute matching for clinical decision making in clinical conditions with aspecific symptoms: the example of syncope” for publication in PLOS ONE (PONE-D-19-19894R1). We would also like to thank you and the reviewer for the thoughtful comments: we have revised the manuscript accordingly and we believe that this process has improved its quality.

Please find attached a marked-up copy of the manuscript that highlights changes made to the original version and an unmarked version of the revised paper without tracked changes, together with point-by-point responses to the points raised during the review process. 

Giovanni Casazza, on behalf of all Co-Authors

Here are our responses to the specific points raised during the review process.

Additional Editor Comments (if provided):

Authors have replied to comments raised in the previous review. Although they agreed in principle with the comments, only minor changes have been made throughout the text. More substantial changes are needed to improve the manuscript. The main idea behind this manuscript should be that the presented methodlogy is interesting and promising and a pilot has been conducted to show this. A larger study is clearly needed to validate the method, either looking at syncope or other conditions. In other words, findings are hypothesis generating rather than conclusive. In this regards, authors should make very clear what their next step will be.

We appreciate the editor’s suggestion. We tried to further clarify that this is a pilot study and that studies on much larger datasets and comparing this method to the traditionally used methods for risk prediction are needed. We added what the next step will be, namely the application of AM to larger datasets through the use of administrative data in conditions in which there is less clinical heterogeneity and to compare AM with the traditional risk stratification tools.

The manuscript has been sent for additional review and comments of Reviewer#2 which are now available need to be carefully addressed in order to meet requirement for publication in PLOS ONE.

Reviewers' comments:

Reviewer's Responses to Questions

Comments to the Author

1. If the authors have adequately addressed your comments raised in a previous round of review and you feel that this manuscript is now acceptable for publication, you may indicate that here to bypass the “Comments to the Author” section, enter your conflict of interest statement in the “Confidential to Editor” section, and submit your "Accept" recommendation.

Reviewer #1: All comments have been addressed

2. Is the manuscript technically sound, and do the data support the conclusions?

Reviewer #1: Partly

3. Has the statistical analysis been performed appropriately and rigorously?

Reviewer #1: No

We appreciate the reviewer’s comment. We have now added logistic regression and updated the methods paragraph.

4. Have the authors made all data underlying the findings in their manuscript fully available?

Reviewer #1: No

We have indicated that data from this study are available upon request. The database analyzed during the current study involves individual-level patients’ data from 5 studies (Costantino G – JACC 2008; Reed MJ – JACC 2010; Del Rosso A – Heart 2008; Quinn JV – Ann Emerg Med 2004; Sun BC – Ann Emerg Med 2007). Public access to the 5 databases is closed and permission to use data for this study had been obtained from each participating center in occasion of the database creation (Costantino G, Am J Med 2014;127:1126.e13-25). Therefore, the authors of the current paper do not have the permission to share de-identified data, which should be asked to the IRB of each corresponding author’s institution.

5. Is the manuscript presented in an intelligible fashion and written in standard English?

Reviewer #1: Yes

6. Review Comments to the Author

Reviewer #1: The authors quote the following sentence in the conclusions: "our study shows that the AM method could be used to predict the risk of adverse events in clinical practice". However, the paper makes no systematic comparison with the state of the art to support this view. Without this comparison the paper turns out to be nothing more than an exercise in style.

Several times the authors have been asked for this by the reviewers, with an answer "we did not perform a formal comparison between AM and clinical judgement. Indeed, the purpose of the study was to describe how the AM method could work in conditions like syncope" and "The purpose of this study was not to make a comparison between AM and the traditional prediction methods. Indeed, we cannot state that AM works better, because a formal comparison would require thousands of patients in the reference database for AM to work. As stated in the study aim, we only wanted to test how AM could work with a real life example and to show that it could allow a different approach". These answers, from my personal point of view, indicate a lack of attention in the normal scientific approach in which, if a method is to be shown, all the specifications for comparison with the other methods must also be provided.

We appreciate the opportunity to clarify. As it is the first study on the use of AM in the context of decision making in clinical conditions with aspecific symptoms, this study is not intended to show that AM works better or worse than logistic regression or clinical judgement, but only to assess its potential applications. Indeed, the intrinsic characteristics of AM and regression make it unlikely that a method based on data such as AM could work better than a method based on a model derived from a small dataset. Therefore, we felt that a systematic comparison with the state of the art would not be appropriate at this stage. However, as the editor and the reviewers deemed it important, we added a comparison with a model based on logistic regression and we are open to any suggestions to improve such comparison. 

Please, provide:

- variable selection criterion and cut-points justification

We appreciate the opportunity to clarify. The database we used was collected for different purposes and we used as predictors the eight variables in common between the original datasets with no a priori decision on the number of predictors to be selected. We acknowledged this in the limitations of the study.

- prediction accuracy measure, in order to enable the reader to judge the method.

We appreciate the reviewer’s suggestion. We added an assessment of the overall diagnostic performance of both multivariate logistic regression and AM with ROC curves and their AUC.

---

## [Editor Report · Decision Letter 2]

23 Jan 2020

Personalized risk stratification through attribute matching for clinical decision making in clinical conditions with aspecific symptoms: the example of syncope

PONE-D-19-19894R2

Dear Dr. Casazza,

We are pleased to inform you that your manuscript has been judged scientifically suitable for publication and will be formally accepted for publication once it complies with all outstanding technical requirements.

With kind regards,

Sandro Pasquali, M.D., Ph.D.

Academic Editor

PLOS ONE

Additional Editor Comments (optional):

Also this time the reviewers comments are well adressed. The manuscript now offers a more balanced view compared to previous versions and is more formally robust also from a methodological viewpoint (i.e. addiction of a logistic regression analysis and changes in discussion/conclusions).

---

## [Editor Report · Acceptance letter]

21 Feb 2020

PONE-D-19-19894R2 

Personalized risk stratification through attribute matching for clinical decision making in clinical conditions with aspecific symptoms: the example of syncope 

Dear Dr. Casazza:

I am pleased to inform you that your manuscript has been deemed suitable for publication in PLOS ONE. Congratulations! Your manuscript is now with our production department. 

With kind regards,

on behalf of

Dr. Sandro Pasquali 

Academic Editor

PLOS ONE